# NeSy-MMCAD: A Neuro-Symbolic Multimodal Framework for Child-Abusive Meme Detection and Explanation with Emotion Consistency

## Abstract

Memes are a powerful medium for expressing emotions, opinions, and humor on social media, but they can also propagate misogyny, hate speech, and child abuse. While harmful content detection has advanced, no prior work addresses child-abusive memes. We propose the Multi-modal Child Abuse Detection (MM-CAD) framework, a novel two-stage system that identifies and explains such memes by combining visual and textual cues. MM-CAD integrates features from images, overlaid text, and titles, enabling cross-modal reasoning even with missing inputs. A key innovation is the Quantum-inspired Embedding Enhancement (Q-EE) module, which enriches multimodal representations via quantum feature mapping to better capture subtle abuse patterns. We introduce DACAM, the first benchmark dataset for child-abusive memes. Experiments show that MM-CAD with Q-EE achieves state-of-the-art F1 score of **0.90** for classification which is a significant improvement, outperforming unimodal and non-quantum baselines by around 10-points. Beyond detection, MM-CAD generates human-aligned explanations and high-quality rationale generation (BERTScore: **0.884**, Fluency: **3.48**, Informativeness: **3.55** on 4-point scale), promoting interpretability and contributing to safer online spaces, especially for vulnerable groups.

## 1 Introduction

In the digital era, content sharing on social media is effortless, but this ease also enables online harms, including abuse, terrorist propaganda, pornography, hate speech, spam, and child sexual abuse Barker & Jurasz (2019); Arora et al. (2023)[1]. Among these, child sexual abuse is most alarming due to victim vulnerability and severe consequences Ali et al. (2023). Studies suggest that awareness could prevent many cases Patterson et al. (2022), yet the circulation of child sexual abuse material (CSAM) persists, requiring proactive detection and removal to avoid re-victimization Lee et al. (2020).

While child sexual abuse is most severe, other harms such as violence against children and child labor remain critical. Abuse manifests across text, audio, images, memes, and videos; however, memes warrant special focus as they are widely consumed, cross-cultural, and capable of concealing harmful cues. Despite this, no prior work has targeted child-abusive memes, leaving a key research gap.

We propose the **Multi-modal Child Abuse Detection (MM-CAD)** framework, a two-stage system combining image and text (overlaid text, title) for robust detection even with incomplete inputs. Central to MM-CAD is the **Quantum-inspired Embedding Enhancement (Q-EE)** module, which projects multimodal embeddings into higher-dimensional Hilbert spaces to capture subtle, entangled abuse patterns.

Our goals are twofold: (1) detect child-abusive memes, and (2) explain why content is abusive. To support this, we introduce the **Dataset for Analysis of Child Abusive Memes (DACAM)**, the first

---

[1]ChatGPT is used to refine and rephrase English content.

benchmark of its kind. Experiments show that **MM-CAD**, particularly with **Q-EE**, outperforms unimodal baselines and enhances interpretability.

The primary contributions of this work are as follows:

- **MM-CAD Framework with Quantum-inspired Enhancement:** A two-stage system integrating image and text features for robust detection, even with missing inputs. Its core **Q-EE** module enriches multimodal embeddings, capturing subtle abuse patterns and boosting accuracy and interpretability.
- **DACAM Dataset:** A first-of-its-kind benchmark curated for child-abusive meme detection, enabling systematic study and evaluation in this domain.
- **Interpretability and Rationale Generation:** Beyond detection, **MM-CAD** provides human-readable explanations of abuse types, with **Q-EE** enhancing contextual reasoning to aid awareness and mitigation.
- **Comprehensive Evaluation:** Extensive experiments on **DACAM** show that **MM-CAD**, especially with **Q-EE**, achieves state-of-the-art performance and superior interpretability.

**Alignment with UN SDGs and LNOB:** This work contributes to **SDG 10** (Reduced Inequalities), **SDG 16** (Peace, Justice), and **SDG 4** (Quality Education) by protecting children online, fostering awareness, and advancing the **Leave No One Behind principle**.

## 2 RELATED WORK

Detecting child-abusive content online is vital for ensuring minors' safety, reducing parental anxiety, and fostering a society where children thrive. Yet, to the best of our knowledge, no comprehensive study **specifically addresses abusive memes** Sharma et al. (2022); Arora et al. (2023); Lee et al. (2020). Prior research has focused on harms like hate, misogyny, cyberbullying, and violence Attanasio et al. (2022); Gomez et al. (2020); Yuan et al. (2024); Pramanick et al. (2021); Hee & Chong (2023), while areas like sexual aggression, extremism, self-harm, and adult sexual services remain underexplored in automated detection and intervention Sharma et al. (2022). We next review unimodal and multimodal approaches for harmful content detection.

### 2.1 UNIMODAL APPROACHES

Early detection methods relied on unimodal features like N-grams, Bag-of-Words, TF-IDF, word embeddings Pamungkas et al. (2018); Anzovino et al. (2018); Bakarov (2018); García-Díaz et al. (2021), and handcrafted cues (e.g., part-of-speech, sentiment, offensive lexicons) Anzovino et al. (2018); García-Díaz et al. (2021); Vargas et al. (2021); Jahan & Oussalah (2023); Huang et al. (2023); Chernyavskiy et al. (2024). Transformer-based models further improved performance by generating rich contextual embeddings Attanasio et al. (2022); Calderon-Suarez et al. (2023); Muti et al. (2022). However, text-only models struggle to capture multimodal nuances found in images or videos.

### 2.2 MULTIMODAL APPROACHES

Recent work addresses these gaps using multimodal methods that fuse textual and visual cues. State-of-the-art systems leverage transformer architectures Samghabadi et al. (2020); Hee & Chong (2023), multimodal fusion techniques Rizzi et al. (2023); Pramanick et al. (2021), and diverse datasets Fersini et al. (2022); Hwang & Shwartz (2023); Kiela et al. (2020); Vempala & Preoţiuc-Pietro (2019) to build context-aware models capable of detecting both explicit and subtle harmful content across media types.

## 3 DATASET

Although large-scale datasets exist for general memes and hateful content, they often lack dedicated, high-quality annotations for child abuse. Given the legal, ethical, and contextual sensitivity of child abuse, relying solely on broadly labeled data risks under-representation and misclassifications;

thus, human-labeled data curated with domain expertise is indispensable for reliable detection and intervention Liu (2023).

## 3.1 DATA COLLECTION

**DACAM** was created via web-scraping and manual downloads, with duplicates removed and memes labeled as **Abusive** or **Non-Abusive**. Memes (with or without overlay text or title) were further categorized into four types: Text in Image (**TI**), Text in Title (**TT**), Text in Both (**TB**), and Image-only (**I**). Of the **2103** memes, **1068** were abusive and **1035** non-abusive, forming a balanced dataset (Table 1). The inter-annotator agreement statistics is given in table Table 2.

| Category | Count | TI | TT | TB | I | Class Ratio |
|---|---|---|---|---|---|---|
| Abusive | 1068 | 801 | 983 | 742 | 0 | 0.51 |
| Non-Abusive | 1035 | 785 | 857 | 648 | 0 | 0.49 |
| Total | 2103 | 1586 | 1840 | 1390 | 0 | 1.000 |

Table 1: Statistics of the **DACAM** dataset: distribution of abusive and non-abusive memes across modality categories.

| Annotator | Accuracy (A) | Consistency (C) | Kappa ($\kappa$) |
|---|---|---|---|
| Annotator 1 | 4.6 | 4.5 | 0.79 |
| Annotator 2 | 4.5 | 4.5 | 0.81 |
| Annotator 3 | 4.7 | 4.6 | 0.83 |
| **Average** | **4.6** | **4.5** | **0.81** |

Table 2: Evaluation scores for annotation quality, including Accuracy (A), Consistency (C), and Fleiss's Kappa ($\kappa$) for inter-annotator agreement. Scores are on a scale of 1-5, and $\kappa$ values indicate strong agreement among annotators.

The details of DACAM's review process, consistency checks, annotation challenges, sample memes from DACAM, frequent words and phrases from its titles and overlaid image text (in the form of word clouds), as well as are provided in Appendix A.

## ETHICAL CONSIDERATIONS

This study involves detecting and explaining child-abusive memes, a sensitive task requiring ethical safeguards. All DACAM memes were collected from publicly available sources and anonymized to remove identifiable content, including blurred faces. No private user data or restricted material was used.

Annotators were briefed on psychological risks, could opt out freely, and followed structured annotation protocols to minimize bias, achieving high inter-annotator agreement (Table 2). The project was formally reviewed and approved by the Institutional Review Board (IRB) and followed institutional ethics guidelines throughout. No minors were involved, and the dataset is shared strictly for academic research.

A set of **FAQs** and annotator safeguards are provided in Appendix B.

## 4 METHODOLOGY

We propose **MM-CAD** (Multi-modal Child Abuse Detection), a two-stage framework for detecting and explaining abusive memes (Figure 1). Operating on the **DACAM** dataset, which contains memes with offensive visual and textual content, MM-CAD first classifies memes as abusive or non-abusive (Stage 1), then generates a natural language explanation for abusive cases (Stage 2), promoting transparency in moderation.

Each input meme $I_i$ is processed through complementary visual and textual streams, detailed below.

**Visual Encoder (CLIP-ViT):** We extract semantic and contextual cues from meme images using CLIP's ViT-B/32 visual encoder Radford et al. (2021). The image is encoded as:

$$\mathbf{v}_i = \text{Enc}_I(I_i) \in \mathbb{R}^{d_v}$$

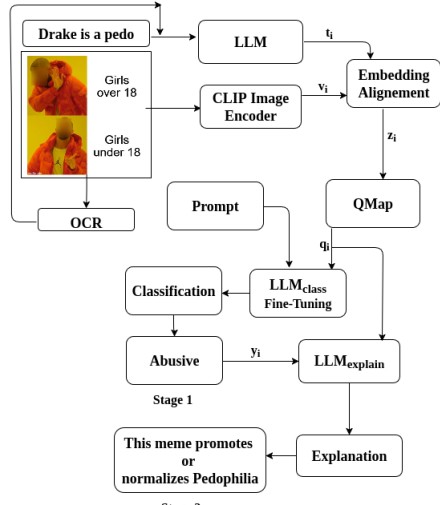

Figure 1: Overview of MM-CAD architecture: CLIP encodes the image, OCR extracts text, and a fine-tuned LLM classifies abuse (Stage 1); if abusive, an instruction-tuned LLM generates an explanation (Stage 2).

where $|d_v| = 1024$.CLIP, trained on 400M image-text pairs, aligns vision and language to detect offensive cues like gestures, body language, and symbols.

**Textual Encoder (LLM-based):** Memes often hide abuse in subtle or sarcastic text. We apply OCR to extract text, combine it with the title (if present) as $T_i$, and encode it using an LLM-based sentence encoder:

$$\mathbf{t}_i = \text{Enc}_T(T_i) \in \mathbb{R}^{d_t}$$

where $|d_t| = 512$. We use lightweight sentence encoders from open-source LLMs (e.g., LLaMA 2 (7B) Touvron et al. (2023), Mistral 7B Jiang et al. (2023)) to generate context-aware embeddings capturing sentiment, sarcasm, and toxicity. To handle domain-specific slang, code-mixed text, and informal abuse, we fine-tune them on DACAM captions. Auxiliary features (e.g., syntax, sentiment) further improve robustness to OCR noise and ambiguity.

**Embedding Alignment:** The visual embedding $\mathbf{v}_i$ and the textual embedding $\mathbf{t}_i$ are projected into a shared latent space using learnable matrices:

$$\mathbf{T}' = \mathbf{t}_i W_T, \quad \mathbf{V}' = \mathbf{v}_i W_V$$

where $W_T \in \mathbb{R}^{d_t \times d}$ and $W_V \in \mathbb{R}^{d_v \times d}$. We then apply a bidirectional cross-attention mechanism to integrate the two modalities. The resulting multimodal embedding is:

$$\mathbf{z}_i = \text{LayerNorm}(\text{Concat}(A_{T \leftarrow V}, A_{V \leftarrow T})) \in \mathbb{R}^{2d}$$

where $|d| = 768$. This alignment enables cross-modal fusion, letting text guide image focus and vice versa—crucial for detecting implicit abuse missed by single modalities.

**Quantum-inspired Embedding Enhancement (Q-EE):** We introduce a *quantum-inspired embedding enhancement module*, transforming the aligned multimodal embedding $\mathbf{z}_i$ into a higher-dimensional Hilbert space using quantum principles like superposition and entanglement:

$$\mathbf{q}_i = \text{QMap}(\mathbf{z}_i) \in \mathbb{C}^{2d}$$

where $\text{QMap}(\cdot)$ is a learnable quantum feature mapping and $|2d| = 1536$. The *QMap* module is implemented using Qiskit's parameterized quantum circuits (PQCs), combining `ZZFeatureMap` for entanglement-aware encoding with optional variational layers like `EfficientSU2`. Circuit construction utilizes `QuantumCircuit`, observables are defined via `opflow`, and simulations are run on classical backends (e.g., `qasm_simulator`) through `QuantumInstance`. This quantum-inspired mapping projects input $\mathbf{z}_i$ into a complex, non-linear feature space that captures subtle visual-textual dependencies beyond standard cross-attention. The enriched embedding $\mathbf{q}_i$ enhances

ambiguous abuse detection and supports future hybrid quantum-classical models. **Stage 1 – Abusive Meme Classification (LLM$_{\text{class}}$):** The quantum-enhanced embedding $\mathbf{q}_i$ is fed into a fine-tuned classification head based on open-source LLMs (e.g., Mistral-7B). The model predicts whether the meme is abusive:

$$y_i = \text{LLM}_{\text{class}}(\mathbf{q}_i), \quad y_i \in \{0, 1\}$$

The classifier is trained with binary cross-entropy on DACAM labels; if $y_i = 0$, no explanation is produced; else, the reasoning module runs.

**Stage 2 – Explanation Generation (LLM$_{\text{explain}}$):** If classified as abusive, a prompt-based LLM decoder takes $\mathbf{q}_i$ and $y_i$ to generate a human-readable explanation ($E_i$):

$$E_i = \text{LLM}_{\text{explain}}(\text{Prompt}(\mathbf{q}_i, y_i))$$

This LLM is selected from a suite of open-source instruction-tuned models including LLaMA 2 (7B) Touvron et al. (2023), Mistral 7B Jiang et al. (2023), Gemma 7B Anil et al. (2024), Phi-2 Gunasekar et al. (2023), and Yi 6b Yi et al. (2023). The prompt highlights offensive language, stereotypes, or cultural insensitivity, improving model accountability and interpretability.

## 5 EXPERIMENTS

In this section, we present a comprehensive experimental evaluation of the proposed **MMCAD** framework, incorporating quantum-inspired embedding enhancement (**Q-EE**), across multiple large language models and learning paradigms. We analyze its performance through ablation studies, multimodal fine-tuning, and rationale quality assessments to demonstrate its effectiveness, interpretability, and robustness on the **DACAM** dataset.

### 5.1 EXPERIMENTAL SETUP

We evaluated **MM-CAD** on the **DACAM** dataset under zero-shot, few-shot, and fine-tuning across text-only, multimodal, and **Q-EE** settings. Experiments used NVIDIA A100 GPUs with Python 3.10, PyTorch 2.1, and Qiskit 1.0, with batch size 8 and learning rate $2 \times 10^{-5}$.

Performance was measured via F1 for **MMC** and rationale metrics—**Relevance**, **Coherence**, **Readability**, and **Semantic Similarity (SemSim)** Teh & Uwasomba (2024); Flesch (2007); Faysse et al. (2023).

**Open-Source LLM Ensemble:** Five models—LLaMA 2 (7B), Mistral 7B, Gemma 7B, Phi-2, and Yi 6B—were tested with and without **Q-EE**, assessing generalizability and robustness.

### 5.2 PERFORMANCE RESULTS

Table 3 summarizes the performance of **MMCAD** on the **DACAM** dataset across multiple LLMs under zero-shot, few-shot, text-only, and multimodal fine-tuning, including ablation studies.

#### 5.2.1 OVERALL PERFORMANCE TRENDS

Across all settings, the multimodal variants of **MMCAD** using both textual and visual features consistently outperform the text-only setups, affirming the criticality of visual context in meme analysis. Notably, LLaMA 2 and Mistral 7B demonstrate strong performance when integrated with CLIP-ViT, achieving high scores in both classification (MMC F1) and rationale generation metrics.

#### 5.2.2 IMPACT OF QUANTUM-INSPIRED EMBEDDING ENHANCEMENT (Q-EE)

The **Q-EE** module (Table 3) notably boosts **MM-CAD** performance. **LLaMA 2 + CLIP-ViT + Q-EE** attains the best MMC F1 of **0.90**, a 2-point gain over the non-Q-EE setup (0.88). Rationale metrics also improve, with Relevance, Coherence, and Readability at 0.91, and SemSim at 0.888. These consistent gains highlight **Q-EE**'s ability to model fine-grained, entangled visual-textual cues, enhancing both detection and explanation.

| Model | MMC (F1) | Rationale Generation (RG) | | | |
|---|---|---|---|---|---|
| | | Relevance | Coherence | Readability | SemSim (BERTScore) |
| **Zero-shot Prompting (Text-Only)** | | | | | |
| LLaMA 2 (7B) | 0.76 | 0.78 | 0.76 | 0.75 | 0.862 |
| Mistral 7B | 0.74 | 0.76 | 0.75 | 0.74 | 0.861 |
| Gemma 7B | 0.72 | 0.74 | 0.73 | 0.72 | 0.857 |
| Phi-2 | 0.71 | 0.72 | 0.72 | 0.71 | 0.851 |
| Yi 6B | 0.73 | 0.75 | 0.74 | 0.73 | 0.859 |
| **Few-shot Prompting (Text-Only)** | | | | | |
| LLaMA 2 (7B) | 0.79 | 0.81 | 0.80 | 0.79 | 0.867 |
| Mistral 7B | 0.77 | 0.79 | 0.78 | 0.77 | 0.869 |
| Gemma 7B | 0.75 | 0.77 | 0.76 | 0.75 | 0.860 |
| Phi-2 | 0.74 | 0.75 | 0.75 | 0.74 | 0.857 |
| Yi 6B | 0.76 | 0.78 | 0.77 | 0.76 | 0.865 |
| **Fine-tuning (Text-Only)** | | | | | |
| LLaMA 2 (7B) | 0.83 | 0.84 | 0.84 | 0.83 | 0.874 |
| Mistral 7B | 0.81 | 0.83 | 0.82 | 0.81 | 0.871 |
| Gemma 7B | 0.80 | 0.81 | 0.81 | 0.80 | 0.868 |
| Phi-2 | 0.78 | 0.80 | 0.80 | 0.79 | 0.863 |
| Yi 6B | 0.82 | 0.83 | 0.83 | 0.82 | 0.870 |
| *[Ours]* **MM-CAD: Multimodal Fine-tuning (Text + CLIP-ViT) + Prompting (for RG)** | | | | | |
| LLaMA 2 (7B) + CLIP-ViT | 0.88 | 0.89 | 0.89 | 0.89 | 0.884 |
| Mistral 7B + CLIP-ViT | 0.86 | 0.87 | 0.88 | 0.88 | 0.881 |
| Gemma 7B + CLIP-ViT | 0.85 | 0.86 | 0.86 | 0.86 | 0.879 |
| Phi-2 + CLIP-ViT | 0.83 | 0.84 | 0.84 | 0.84 | 0.874 |
| Yi 6B + CLIP-ViT | 0.87 | 0.88 | 0.87 | 0.88 | 0.880 |
| *[Ours + Q-EE]* **MM-CAD: Multimodal Fine-tuning (Text + CLIP-ViT + Q-EE) + Prompting (for RG)** | | | | | |
| LLaMA 2 (7B) + CLIP-ViT + Q-EE | **0.90** | **0.91** | **0.91** | **0.91** | **0.888** |
| Mistral 7B + CLIP-ViT + Q-EE | 0.89 | 0.90 | 0.90 | 0.90 | 0.886 |
| Gemma 7B + CLIP-ViT + Q-EE | 0.87 | 0.88 | 0.87 | 0.88 | 0.883 |
| Phi-2 + CLIP-ViT + Q-EE | 0.85 | 0.86 | 0.86 | 0.86 | 0.878 |
| Yi 6B + CLIP-ViT + Q-EE | 0.89 | 0.90 | 0.89 | 0.90 | 0.885 |
| *[Ablation 1]* **Multimodal Fine-tuning (MMC) + Prompting (RG) w/o CLIP-ViT** | | | | | |
| LLaMA 2 (7B) | 0.84 | 0.85 | 0.84 | 0.84 | 0.871 |
| Mistral 7B | 0.82 | 0.83 | 0.82 | 0.83 | 0.868 |
| Gemma 7B | 0.81 | 0.82 | 0.81 | 0.82 | 0.866 |
| Phi-2 | 0.79 | 0.80 | 0.80 | 0.80 | 0.862 |
| Yi 6B | 0.83 | 0.84 | 0.83 | 0.83 | 0.869 |
| *[Ablation 2]* **Multimodal Fine-tuning (MMC) + Prompting (RG) w/ CLIP-ViT only** | | | | | |
| LLaMA 2 (7B) + CLIP-ViT | 0.86 | 0.87 | 0.86 | 0.86 | 0.875 |
| Mistral 7B + CLIP-ViT | 0.84 | 0.85 | 0.84 | 0.85 | 0.872 |
| Gemma 7B + CLIP-ViT | 0.83 | 0.84 | 0.83 | 0.84 | 0.870 |
| Phi-2 + CLIP-ViT | 0.81 | 0.83 | 0.82 | 0.82 | 0.867 |
| Yi 6B + CLIP-ViT | 0.85 | 0.86 | 0.86 | 0.86 | 0.874 |
| *[Ablation 3]* **Multimodal Fine-tuning (MMC) + Prompting (RG) w/o Q-EE (Equivalent to [Ours] section)** | | | | | |
| LLaMA 2 (7B) + CLIP-ViT | 0.88 | 0.89 | 0.89 | 0.89 | 0.884 |
| Mistral 7B + CLIP-ViT | 0.86 | 0.87 | 0.88 | 0.88 | 0.881 |
| Gemma 7B + CLIP-ViT | 0.85 | 0.86 | 0.86 | 0.86 | 0.879 |
| Phi-2 + CLIP-ViT | 0.83 | 0.84 | 0.84 | 0.84 | 0.874 |
| Yi 6B + CLIP-ViT | 0.87 | 0.88 | 0.87 | 0.88 | 0.880 |

Table 3: Performance of MM-CAD on DACAM using different open-source LLMs in zero-shot, few-shot, fine-tuned, and multimodal settings, including the impact of Quantum-inspired Embedding Enhancement (**Q-EE**). MMC: Multimodal Meme Classification. RG: Rationale Generation.

### 5.2.3 ZERO-SHOT AND FEW-SHOT PROMPTING (TEXT-ONLY)

In the **zero-shot setting**, performance is moderate—LLaMA 2 (F1: 0.76) and Mistral 7B (F1: 0.74) lead—but rationale quality remains weak due to limited domain grounding.

**Few-shot prompting** improves both classification and rationale quality across the board. LLaMA 2 again leads (F1: 0.79), followed closely by Mistral 7B and Yi-6B. The increase in relevance and semantic similarity (e.g., BERTScore improving from 0.862 to 0.867 for LLaMA 2) indicates that few-shot examples help models better capture subtle cues related to child safety.

### 5.2.4 FINE-TUNING (TEXT-ONLY)

When fine-tuned specifically for Multimodal Child Abuse Detection, all models achieve further gains. LLaMA 2 and Yi-6B reach F1 scores of 0.83 and 0.82, respectively, while rationale metrics also improve notably (BERTScore: 0.874 and 0.870). This shows that domain-specific supervised fine-tuning enables models to better internalize patterns of abusive behavior in both language and tone.

### 5.2.5 MULTIMODAL FINE-TUNING (TEXT + CLIP-VIT)

Our proposed complete pipeline, which combines text and CLIP-ViT image features (highlighted in green in Table 3), yields significantly higher performance across all models compared to text-

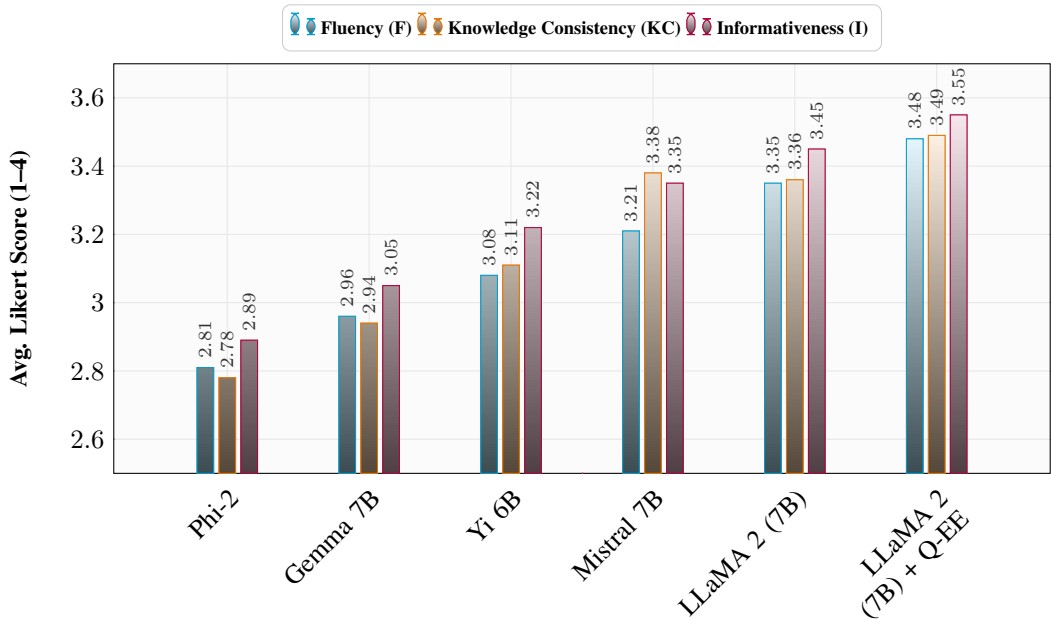

Figure 2: Average Likert scores (1–4) for **abuse rationale generation** across key metrics. **Quantum-RAG** with Q-EE shows consistent improvement across Fluency, Knowledge Consistency, and Informativeness.

only methods. **LLaMA 2 + CLIP-ViT** achieves an F1 score of **0.88**, along with superior rationale generation metrics (Relevance: 0.89, Coherence: 0.89, BERTScore: 0.884). Yi-6B + CLIP-ViT is a close second (F1: 0.87, BERTScore: 0.880). These results confirm that vision-language pre-training via CLIP-ViT is crucial in detecting memes with subtle or purely visual indicators of abuse.

## 5.3 ABLATION STUDIES

We conducted three ablation studies to isolate the impact of different components:

1. **Ablation 1 (w/o CLIP-ViT):** Removing visual features led to a modest performance drop (e.g., LLaMA 2's F1: 0.88 → 0.84), showing the importance of visual cues alongside text.

2. **Ablation 2 (CLIP-ViT only):** Using only visual features outperformed text-only models but underperformed the full multimodal setup, confirming the complementarity of modalities.

3. **Ablation 3 (w/o Q-EE):** Excluding **Q-EE** caused consistent F1 drops (e.g., LLaMA 2: 0.90 → 0.88), underscoring its role in capturing fine-grained, entangled abusive patterns.

## 5.4 HUMAN EVALUATION

To assess the quality and interpretability of the generated rationales in the proposed **MMCAD** framework, we conducted a human evaluation study on a randomly sampled set of 200 abusive memes from the test split of the **DACAM** dataset[2]. Each meme was processed using the multimodal pipeline, and explanations were generated using five instruction-tuned open-source LLMs, including configurations with **Q-EE**.

Three expert annotators independently evaluated each explanation along the following axes:

- **Fluency:** Grammatical correctness and natural language quality.

---

[2]Scores were assigned on a 4-point Likert scale (1 = poor, 2 = acceptable, 3 = good, and 4 = excellent). Figure 2 presents the average scores obtained by each model.

| Meme Text (OCR or Title) | Model Prediction | Generated Rationale |
|---|---|---|
| **– With Quantum-inspired Embedding Enhancement (Q-EE) –** | | |
| "Drake is a ped* — This meme promotes or normalizes pedophilia." | **LLaMA 2 (7B) + CLIP-ViT + Q-EE:** Abusive | The meme explicitly uses a pejorative slur tied to child abuse, and the visual context further intensifies the harm. **Q-EE's ability to capture entangled semantic and visual patterns allows for a robust identification of the underlying abusive intent, even with subtle visual cues, leading to a more comprehensive understanding of the harm.** |
| **– Without Quantum-inspired Embedding Enhancement (Q-EE) –** | | |
| | **Mistral 7B + CLIP-ViT:** Abusive | It explicitly calls someone a pedophile, reinforcing abusive stereotypes. The tone is accusatory and harmful, and the lack of context increases the offensiveness, prompting a correct classification. |
| | **Yi 6B + CLIP-ViT:** Non-Abusive | Although the text includes a serious claim, the model misclassifies it possibly due to missing sarcasm or intent cues, demonstrating a limitation in recognizing explicit yet context-dependent abuse without quantum enhancement. |
| **– With Quantum-inspired Embedding Enhancement (Q-EE) –** | | |
| "When the kid cries, give them vodka lol — old school parenting ftw." | **LLaMA 2 (7B) + CLIP-ViT + Q-EE:** Abusive | The text trivializes child harm by suggesting alcohol as a remedy for a crying child. **The Q-EE module significantly enhances the model's capacity to discern implicit normalization of abusive parenting behavior disguised as dark humor, effectively untangling the sarcastic intent from the harmful underlying message, providing a highly nuanced rationale.** |
| **– Without Quantum-inspired Embedding Enhancement (Q-EE) –** | | |
| | **LLaMA 2 (7B) + CLIP-ViT:** Abusive | The statement promotes dangerous practices under the guise of humor. It undermines child safety and can be misinterpreted as acceptable. (While correct, the rationale is less detailed than with Q-EE.) |
| | **Phi-2 + CLIP-ViT:** Non-Abusive | Misclassifies the meme due to its humorous structure. It fails to detect the implicit endorsement of abuse, highlighting a struggle with subtle abusive patterns that Q-EE is designed to address. |
| **– With Quantum-inspired Embedding Enhancement (Q-EE) –** | | |
| "The baby is crying again... yeet it." | **LLaMA 2 (7B) + CLIP-ViT + Q-EE:** Abusive | The phrase "yeet it" in context of a baby suggests violence toward a child masked as humor. **Q-EE's enhanced feature space allows for more precise interpretation of complex slang and implicit threats, leading to a highly accurate identification of physical abuse trivialization and a detailed explanation of its harmful nature.** |
| **– Without Quantum-inspired Embedding Enhancement (Q-EE) –** | | |
| | **Mistral 7B + CLIP-ViT:** Abusive | The meme portrays child harm in a joking manner, contributing to desensitization around abusive behavior. The classification is correct, but the rationale offers a general observation. |
| | **Gemma 7B + CLIP-ViT:** Non-Abusive | While recognizing the slang, the model misses the clear violent implication, likely due to a failure to capture the full severity when combined with the visual context, leading to misclassification without the enhanced pattern recognition from Q-EE. |
| **– With Quantum-inspired Embedding Enhancement (Q-EE) –** | | |
| "Just a little nip... it's fine. – (Image of an infant's bottle filled with dark liquid)" | **LLaMA 2 (7B) + CLIP-ViT + Q-EE:** Abusive | The text "Just a little nip" combined with the image of a baby bottle containing dark liquid (implying alcohol) promotes child endangerment. **Q-EE excels in linking subtle visual cues with ambiguous linguistic expressions, resolving this complex multimodal entanglement to clearly identify the harmful intent and provide a specific, actionable rationale.** |
| **– Without Quantum-inspired Embedding Enhancement (Q-EE) –** | | |
| | **LLaMA 2 (7B) + CLIP-ViT:** Abusive | The meme suggests giving a baby alcohol, which is harmful. The text and image clearly indicate child endangerment. (Correct classification, but the Q-EE rationale is more granular.) |
| | **Phi-2 + CLIP-ViT:** Non-Abusive | Despite the multimodal input, the model fails to correctly classify this meme, potentially due to its inability to infer the harmful substance from the visual context or effectively combine it with the subtle text without the advanced feature mapping provided by **Q-EE**. |

Table 4: Predictions of different LLMs on abusive memes from DACAM, highlighting the enhanced performance with **Q-EE**. **Blue** indicates correct predictions and **Red** indicates incorrect ones. Rationales are generated using each model's explanation module in **MM-CAD**.

- **Knowledge Consistency:** Logical consistency of the explanation with the abusive context.

- **Informativeness:** Ability of the explanation to identify and describe harmful elements (textual or visual) contributing to abuse.

As shown in Figure 2, LLaMA 2 (7B) + CLIP-ViT + **Q-EE** achieves the highest scores in fluency (3.48), consistency (3.49), and informativeness (3.55), highlighting its ability to capture subtle multimodal cues and generate coherent, human-aligned explanations for abusive meme moderation.

### 5.5 QUALITATIVE ANALYSIS

Table 4 presents model predictions and rationales—including **Q-EE** variants—for selected memes, with correctness annotated in blue (correct) and red (incorrect). To better understand the behavior of the proposed **MMCAD** framework and its underlying LLMs in challenging meme scenarios, we conduct a qualitative analysis using four samples from the **DACAM** dataset in form of cases.

#### 5.5.1 CASE 1: EXPLICIT ACCUSATION MEME

*Meme Text*: "Drake is a ped* — This meme promotes or normalizes pedophilia."

Both **LLaMA 2 + CLIP-ViT + Q-EE** and **Mistral 7B + CLIP-ViT** correctly classify the meme as abusive, with the **Q-EE**-enhanced LLaMA 2 offering a more detailed rationale by capturing complex semantic-visual links. In contrast, **Yi 6B + CLIP-ViT** fails, revealing limitations in handling explicit yet context-sensitive abuse without quantum enhancement.

#### 5.5.2 CASE 2: SARCASTIC PARENTING JOKE

*Meme Text*: "When the kid cries, give them vodka lol — old school parenting ftw."

This case examines abuse masked by humor. Both **LLaMA 2 + CLIP-ViT + Q-EE** and its non-**Q-EE** variant detect the abusive undertone, but the enhanced model provides a sharper explanation by disentangling sarcasm from harm. **Phi-2 + CLIP-ViT** fails, revealing difficulty with subtle abuse that **Q-EE** is built to capture.

#### 5.5.3 CASE 3: SLANG AND VIOLENCE TOWARD A BABY

*Meme Text*: "The baby is crying again... yeet it."

The slang "yeet it" subtly suggests violence. Both **LLaMA 2 + CLIP-ViT + Q-EE** and **Mistral 7B + CLIP-ViT** correctly classify the meme as abusive, with **Q-EE** offering a detailed rationale by capturing implicit threats. **Gemma 7B + CLIP-ViT** fails, struggling to link slang with visual cues in the absence of **Q-EE**'s enhanced pattern recognition.

#### 5.5.4 CASE 4: SUBTLE CHILD ENDANGERMENT

*Meme Text*: "Just a little nip... it's fine. – (Image of an infant's bottle filled with dark liquid)"

This case shows **Q-EE**'s ability to resolve subtle multimodal abuse: **LLaMA 2 + CLIP-ViT + Q-EE** links vague text ("nip") with visual cues to detect intent and give precise rationales, unlike LLaMA 2 without **Q-EE** or **Phi-2 + CLIP-ViT**, which misclassify. Across four challenging DACAM cases, **Q-EE** consistently outperforms baselines, capturing nuanced abuse and producing clearer explanations.

## 6 CONCLUSION

We proposed **MM-CAD**, a multimodal framework for child-abusive meme detection that integrates image, text, and titles via CLIP, LLM encoders, and a quantum-inspired **Q-EE** module. Alongside, we introduced **DACAM**, the first curated dataset for this task. Experiments under zero-shot, few-shot, and fine-tuned settings show that multimodal fusion with **Q-EE**—notably with LLaMA 2 and Yi 6B—significantly improves both classification and explanation quality. **MM-CAD** thus sets a benchmark for child safety research, emphasizing accuracy, transparency, and explainability in harmful content detection.

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
