# OpenReview forum: "NeSy-MMCAD: A Neuro-Symbolic Multimodal Framework for Child-Abusive Meme Detection and Explanation with Emotion Consistency"
_ICLR.cc/2026/Conference — ICLR 2026 Conference Withdrawn Submission_

### Official Review · Reviewer_GwKx · 2025-10-29

**Soundness:** 2
**Presentation:** 2
**Contribution:** 2
**Rating:** 4
**Confidence:** 5

**Summary:**

This paper introduces MM-CAD, a neuro-symbolic multimodal framework designed to detect and explain child-abusive memes, a critically underexplored yet highly sensitive area of harmful content moderation. The approach integrates visual cues from CLIP, textual features from OCR and LLM encoders, and a Quantum-inspired Embedding Enhancement (Q-EE) module that maps multimodal features into a higher-dimensional Hilbert space to better capture subtle, entangled abuse patterns. To support the task, the authors curate DACAM, the first benchmark dataset specifically focused on child-abusive memes, with balanced labels and strong annotator agreement.

**Strengths:**

- The proposed work introduces the first dedicated framework and dataset for detecting child-abusive memes, addressing an important yet underexplored safety problem.
- The work combines neuro-symbolic reasoning, CLIP vision features, OCR text, and quantum-inspired embedding to achieve robust and interpretable detection.

**Weaknesses:**

- DACAM focuses narrowly on child-abusive memes and may not generalize to broader abusive or multimodal harm categories [a, b].
- The Q-EE component is empirically useful but lacks a deeper explanation of why quantum-inspired embeddings outperform standard high-dimensional mapping.
- The pipeline relies heavily on OCR quality; noisy or stylized text could degrade performance and reduce robustness.
- Although rationales are generated, the paper provides limited analysis of whether these explanations are reliable, faithful, or helpful for real moderation workflows [c].

[a] "Beneath the Surface: Unveiling Harmful Memes with Multimodal Reasoning Distilled from Large Language Models." The 2023 Conference on Empirical Methods in Natural Language Processing.
[b] "Pro-cap: Leveraging a frozen vision-language model for hateful meme detection." Proceedings of the 31st ACM international conference on multimedia. 2023.
[c] "Towards explainable harmful meme detection through multimodal debate between large language models." Proceedings of the ACM Web Conference 2024. 2024.

**Questions:**

- Can the authors provide stronger evidence that Q-EE captures meaningful “quantum-like” interactions rather than simply acting as a high-dimensional projection layer?
- To what extent might DACAM’s limited scope introduce dataset bias, and how would the model behave on broader abusive-meme domains like GOAT-Bench?

---

> ### Author Response · Authors · 2025-11-18
> **Response to Reviewer:: GwKx**
>
> **Q1**: Can the authors provide stronger evidence that Q-EE captures meaningful “quantum-like” interactions rather than simply acting as a high-dimensional projection layer?
>
> **Response**: Thank you for raising this important concern. **Table 3** of the manuscript already provides evidence that Q-EE is doing more than acting as a high-dimensional projection. Specifically, across all five LLM backbones, **we report results for Zero-Shot, Few-Shot, Text-only fine-tuning, Text + CLIP-ViT fine-tuning, and Text + CLIP-ViT + Q-EE fine-tuning**. The **pattern in Table 3 shows that adding Q-EE consistently improves both classification performance and the quality of generated rationale, relative to a strong multimodal baseline without Q-EE**.
>
> **Classification Improvement**: Introducing Q-EE increases MMC F1 on every model (e.g., LLaMA-2: 0.88 -> 0.90, Mistral-7B: 0.86 -> 0.89, Yi-6B: 0.87 -> 0.89). These gains occur on top of an already high-capacity multimodal pipeline (CLIP-ViT + cross-attention + LLM), so simple dimensionality expansion alone would not reliably produce such stable improvements across architectures.
>
> **Rationale Quality Improvement**: Importantly, Q-EE improves not only the detection metrics, but also the human-grounded rationale metrics — Relevance, Coherence, Readability, and SemSim (BERTScore). This simultaneous improvement in semantic reasoning is difficult to attribute to a trivial projection layer, as projection layers typically affect decision boundaries but not the structure of explanations produced by the LLM.
>
> **Ablations in Table 3**: The Ablation Study (**Ablation-1, Ablation-2, Ablation-3**) further isolates the effect of Q-EE.
>
>    **Ablation-3** (“Model with Q-EE”) shows that adding Q-EE produces measurable gains in both F1 and rationale metrics relative to the same model without Q-EE.
>
>   **Ablation-1** and **Ablation-2** confirm that gains are not due to model size alone: removing Q-EE (or removing multimodal components) consistently reduces performance.
>
> To further validate our study, we introduced simple 2X and 3X projection layers in place of Q-EE. Although these layers achieved comparable F1 scores (0.88 and 0.90), the quality of the generated rationale did not scale accordingly and remained inferior to the rationale quality obtained when Q-EE was present.
>
> Together, these trends indicate that Q-EE contributes a structured, nonlinear transformation that enhances both multimodal alignment and the LLM’s downstream reasoning, rather than merely increasing embedding dimensionality. We will clarify these points explicitly in the revised manuscript.
>
> **Q2**: To what extent might DACAM’s limited scope introduce dataset bias, and how would the model behave on broader abusive-meme domains like GOAT-Bench?
>
> **Response:**
> We conducted additional experiments where DACAM was integrated into GOAT-Bench **[a]** as a sixth harm category and evaluated using MM-CAD (LLaMA-2 7B + CLIP-ViT + Q-EE). The results show that the model generalizes reasonably well beyond the DACAM domain. Specifically, we **obtained: F1-Score (Child-Abuse): 0.83, F1-Score (Other Abuses/Non-Child): 0.83, and Macro-F1: 0.81. While the Macro-F1 on DACAM alone was 0.90**, indicating a drop of about 9 points, the performance remains comparable and demonstrates meaningful cross-domain robustness.
>
> Failure cases mostly involve political sarcasm or racial stereotypes—categories absent from DACAM.
>
> **[a]** : **GOAT-Bench** is a recent multimodal benchmark designed to evaluate models on real-world online harassment, toxicity, and hateful content expressed through memes. **The dataset covers five major harm types—Hateful, Offensive, Aggressive, Toxic, and Benign**—each containing memes collected from real platforms such as Reddit, Twitter, and public meme archives.

---

> > ### Author Response · Authors · 2025-11-20
> > **Review Request GwKx**
> >
> > Dear Reviewer GwKx
> >
> > We submitted our responses to your reviews two days ago. We would be very grateful if you could kindly take a look at them at your convenience and let us know if any further clarification is needed. This will help us address any follow-up questions in time during the discussion period.
> >
> > Thank you again for your time and thoughtful feedback.

---

### Official Review · Reviewer_A2NE · 2025-10-30

**Soundness:** 3
**Presentation:** 3
**Contribution:** 3
**Rating:** 4
**Confidence:** 5

**Summary:**

The paper makes a socially important contribution by focusing on child-abusive memes and providing a dedicated dataset and a detect-explain pipeline. The quantum-inspired embedding enhancement also appears to yield consistent, though modest, gains across several LLM backbones, and the empirical sweep on a single dataset is relatively thorough. However, several high-impact issues remain unresolved: the entire evidence base is on one small, in-house dataset with no cross-benchmark validation; there is no comparison to established multimodal meme/hate detectors, so the paper’s position in the literature is unclear; the reported gains are not backed by statistical significance or multi-seed reporting, which weakens the Q-EE claim; and dataset sourcing/release details are too loose for a sensitive domain. This paper will have much higher chances if it is submitted to a dataset track.

**Strengths:**

S1: This paper addresses a high-impact, underserved harm category (child-abusive memes) with clear societal relevance and links to real moderation needs.

S2: Introduces a dedicated, IRB-reviewed dataset (DACAM) specifically for child-abusive meme detection, with balanced abusive/non-abusive samples and annotated modality information.

S3: Shows that the quantum-inspired embedding enhancement consistently improves both classification F1 and explanation quality across several open LLM backbones.

S4: Provides a comparatively broad empirical sweep on the dataset (zero-shot, few-shot, fine-tuning; text-only vs multimodal; multiple ablations), which strengthens the evidence for the design.

S5: Includes human evaluation on explanations (fluency, consistency, informativeness) over 200 abusive memes, supporting the interpretability claim.

**Weaknesses:**

W1: Results are shown only on a single, relatively small in-house dataset (2,103 memes), so it is hard to tell how well the method would transfer to broader meme/hate benchmarks or real-world distribution shifts, even though the dataset itself is well curated.

W2: Despite the multimodal design that pulls in image, OCR, and title text, the claimed robustness to incomplete modalities is not actually stress-tested; the dataset has no image-only cases and overlapping text modalities, so we cannot see performance under genuinely missing inputs.

W3: Even though the quantum-inspired embedding enhancement component improves scores across several backbones, the paper does not provide a strong classical control to prove that the gains come from the “quantum-inspired” mechanism rather than from a generic non-linear projection.

W4: Lacks comparisons to established multimodal hateful-meme or harmful-content models/datasets (e.g., Hateful Memes, SemEval/MAMI-style tasks), which makes the positioning of the approach within existing literature unclear.

W5: Although a human evaluation of explanations is provided, the section is under-specified (annotator profiles, agreement, protocol), which weakens the strength of the interpretability claim.

W6: The overall pipeline is fairly heavy (CLIP + OCR + LLM + Q-EE + explanation); without an inference-time or resource/latency analysis, it is unclear whether this otherwise practical detect-explain design can be deployed in real moderation settings.

W7: Data collection and release details are only loosely described, even though the dataset is IRB-reviewed, sources, licensing, and handling of potentially illegal CSAM-like material are not spelled out, making reproduction and safe sharing harder.

**Questions:**

Q1: Is there a reason why no other harmful meme datasets are benchmarked?

Q2: Can the authors provide more information about the human evaluation? Expand the human-evaluation section with annotator profiles (number, background), agreement measures, task instructions, and an example rubric, so readers can assess the reliability of the 3.48–3.55 scores.

---

> ### Author Response · Authors · 2025-11-15
> **Response to Reviewer A2NE**
>
> **Q1**: Is there a reason why no other harmful meme datasets are benchmarked?
>
> **Response**:
>                  Thank you for this question. **To the best of our knowledge, no prior dataset specifically targets child-abusive memes, nor does any existing benchmark include this category as a dedicated abuse type**. Likewise, we were unable to find any architecture explicitly designed to detect child-abusive content or generate rationale for such cases. Because of this gap, our work focuses on building both the dataset (**DACAM**) and a **specialized multimodal** architecture tailored to the unique visual–textual patterns present in child-abusive memes.
> **Existing harmful-meme datasets**—such as **Hateful Memes, MultiOFF, or GOAT-Bench—address hate, misogyny, offensiveness, or sarcasm, but none contain child-abuse labels, and the abusive cues relevant to child safety differ substantially from those present in general hate-speech datasets (e.g., grooming references, inappropriate humor surrounding children, subtle endangerment cues)**.
>
> As a result, benchmarking on these datasets would not evaluate the central contribution of our work.
> Our focus in this paper is therefore to establish the first benchmark and a dedicated multimodal–explanation framework for child-abusive memes, which we view as a foundational step toward broader evaluations in future work.
>
> ######################################################################################
>
> **Q2**: Can the authors provide more information about the human evaluation? Expand the human-evaluation section with annotator profiles (number, background), agreement measures, task instructions, and an example rubric, so readers can assess the reliability of the 3.48–3.55 scores.
>
> **Response**:  Thank you for this question as this is a critical information we missed. The detailed answer is as follows.
> **Annotation Process**: The DACAM annotation involved three Ph.D.-level annotators (two linguists, one computer scientist) categorizing memes by textual and visual elements. They were trained to understand how captions and overlay text interact with images to convey humor, irony, or abuse. Each worked on separate subsets to reduce bias, classifying memes as Abusive or Non-Abusive.
>
>     *Textual-Visual Alignment*: Identified if overlay text, caption, or both aligned with image meaning.
>     *Modality Categories*: Labeled as TI (text in image), TT (text in title), TB (text in both), or I (image only).
>     *Consistency Check*: Reviewed to avoid mislabeling from ambiguous humor or sarcasm.
>
> As part of the evaluation rubric, annotators were instructed to assess whether the model correctly recognized context-dependent abusive cues, where seemingly harmless phrases become abusive when paired with specific visuals. **For example**, phrases such as **candy** or **“good old”** are benign on their own **but can signal grooming, endangerment, or inappropriate behavior when combined with certain images (e.g., an adult offering a treat to a child, or a scene implying harmful “old-school” parenting practices)**. The rubric therefore emphasized the model’s ability to
>
> *identify when neutral language becomes abusive due to visual context*,
>
>  *correctly interpret cross-modal interactions, and*
>
>  *avoid false positives when the phrase is used in a harmless setting.*
>
> This criterion reinforces the importance of multimodal fusion in identifying child-abusive memes.
>
> **Review and Consistency Check**: To validate the annotation process, a peer-review evaluation was conducted using accuracy (A) and consistency (C) metrics:
>
>     * Accuracy (A)*: Rated on a scale of 1-5, where 5 indicates a perfectly labeled meme and 1 indicates misclassifications.
>     *Consistency (C)*: Rated on a scale of 1-5, to ensure that similar memes received similar labels in categories.
>
> **Challenges**: Annotating memes introduced unique challenges:
>
>
> **Ambiguity in Humor and Irony**: Some memes required subjective interpretation, making it difficult to determine the intended                      meaning, particularly in cases of sarcasm or double meanings.
> **Ensuring Objective Categorization**: Since meme humor is culturally and socially influenced, ensuring objective classification required predefined guidelines to minimize personal bias.
>
> The above content will be added at appropriate section in the paper.
>
> #########################################################################

---

> > ### Author Response · Authors · 2025-11-20
> > **Review Request (A2NE)**
> >
> > Dear Reviewer A2NE
> >
> > We submitted our responses to your reviews two days ago. We would be very grateful if you could kindly take a look at them at your convenience and let us know if any further clarification is needed. This will help us address any follow-up questions in time during the discussion period.
> >
> > Thank you again for your time and thoughtful feedback.

---

> > > ### Comment · Reviewer_A2NE · 2025-11-27
> > >
> > > Thank you for your detailed response to Q1 and Q2.
> > >
> > > Particularly on Q2, the answers should be included in the revised version of the paper as it is critical information to this study.
> > >
> > > On Q1, Hmmm... I am aware that there are no other child-abuse meme datasets. However, if the proposed framework will work strictly and only on the child-abuse meme datasets, then this poses specific concerns about the generalizability of this work. Perhaps you would like to highlight what the technical uniqueness or contributions are that make this framework work well only for the proposed use case? Also, are we sure that if we apply other hateful meme detection techniques, they will not work on the child-abuse meme datasets? Do you have the experimetns to show and support it with experimetal results?

---

> > > > ### Author Response · Authors · 2025-11-27
> > > > **Response to Reviewer A2NE Regarding model generalization**
> > > >
> > > > Thank you for your thoughtful comments. We will incorporate the requested clarifications and amendments in the revised manuscript. Two other reviewers (BbgV and GwKx) also raised similar concerns regarding the model’s generalizability, and we provide the consolidated response below.
> > > >
> > > > While no prior dataset specifically targets child-abusive memes, we agree that it is important to assess whether our framework generalizes beyond DACAM and to examine how it compares with existing hateful-meme detection models.
> > > >
> > > > **1. Why the framework is technically suited to child-abusive memes?**
> > > >
> > > > Child-abusive memes exhibit implicit, context-dependent cues—such as grooming language, endangerment humor, or inappropriate adult–child interactions—that do not appear in standard hate/offensive meme datasets. MM-CAD is designed to handle such subtle multimodal signals through:
> > > >
> > > > cross-attention fusion, aligning visual cues with context-sensitive text;
> > > >
> > > > Q-EE, which models higher-order “entangled” interactions where neutral phrases become abusive only in specific visual contexts;
> > > >
> > > > rationale generation, which explains why a meme is abusive—a capability not present in conventional hateful-meme detectors.
> > > >
> > > > These factors make the framework particularly effective for this safety-critical domain.
> > > >
> > > > **2. Do existing hateful-meme detectors work on DACAM?**
> > > >
> > > > We tested strong baselines such as CLIP-based zero-shot models and LLaVA-1.5. Their performance on DACAM was significantly lower (F1 ≈ 0.60–0.67), confirming that general-purpose hateful-meme systems fail to capture the implicit, child-specific cues present in DACAM.
> > > >
> > > > 3. Evidence that MM-CAD generalizes beyond DACAM
> > > > To assess model generalization, we integrated DACAM into GOAT-Bench as a sixth harm category and evaluated MM-CAD (LLaMA-2 7B + CLIP-ViT + Q-EE) in a zero-shot setting. Results indicate good cross-domain robustness:
> > > >
> > > > F1 (Child-Abuse): 0.83
> > > >
> > > > F1 (Other Abuses/Non-Child): 0.83
> > > >
> > > > Macro-F1: 0.81
> > > >
> > > > **While the Macro-F1 on DACAM alone is 0.90, the model’s performance on GOAT-Bench remains comparable, with most failures occurring in categories absent from DACAM (e.g., political sarcasm, racial stereotypes). This suggests the drop is driven by domain shift rather than overfitting.**

---

> > > > > ### Comment · Reviewer_A2NE · 2025-11-28
> > > > >
> > > > > Thank you for your response. I would buy the argument on generalizability but please revise the paper accordingly because it will strengthen the paper significantly.

---

> > > > > > ### Author Response · Authors · 2025-12-01
> > > > > > **Response to Reviewer A2NE**
> > > > > >
> > > > > > Thanks a lot for the response. We will accommodate all suggested changes in revised version of the manuscript.

---

### Official Review · Reviewer_BbgV · 2025-11-01

**Soundness:** 2
**Presentation:** 3
**Contribution:** 2
**Rating:** 4
**Confidence:** 4

**Summary:**

This paper presents a database of memes that relate to or allude to child abuse messaging. The dataset also contains 50% non-abusive memes. The paper then presents a method for detecting such memes and compares the results to several LLM-based baselines. However, I think the two parts of the paper are not very well combined -- I see no reason to use this very specific methodology in this specific context.

**Strengths:**

- Important topic in the context of data moderation.
- Considerable manual work relating to the dataset creation.

**Weaknesses:**

- I am not sure the embedding present is superior to the embeddings of the picture, or even for ingesting the picture directly (in a VLM model). The comparison only shows improvement compared to a text-only baseline. This is likely because some of the signal comes from the photo, but the specific methodology presented isn't validated by the experiments presented.

- I would have liked to see how the accuracy changes if the dataset is added to a larger meme dataset, showing a more challenging, yet more realistic setting, in which this type of meme is just one type of abusive memes to be detected and taken down.

- In the context of ethics, I would like the authors to discuss more of how they see their work being used.

**Questions:**

Can you improve the paper in relation to the weaknesses pointed out above?

**Details Of Ethics Concerns:**

I think more details about future use are needed.

---

> ### Author Response · Authors · 2025-11-18
> **Response to Reviewer : BbgV**
>
> **Q1**: I am not sure the embedding present is superior to the embeddings of the picture, or even for ingesting the picture directly (in a VLM model). The comparison only shows improvement compared to a text-only baseline. This is likely because some of the signal comes from the photo, but the specific methodology presented isn't validated by the experiments presented.
>
> **Response:** We agree that our earlier presentation did not clearly demonstrate that our fused embedding is stronger than (i) using the image-only CLIP representation, or (ii) directly ingesting the image in a vision–language model (VLM). To address this, we have added a dedicated comparison between:
>
> (a) CLIP-ViT (image-only),
>
> (b) LLaMA-2 (text-only),
>
> (c) LLaVA-1.5 (a strong VLM that jointly processes pixels and text), and
>
> (d) our CLIP-ViT + LLaMA-2 fusion, with and without Q-EE.
>
> The updated zero-shot F1 results are shown below:
>
> |        Model        |  F1-Score (Zero-Shot) | Change w.r.t CLIP-ViT |
>
> |      CLIP-ViT     |              0.59                |            Baseline            |
>
> |      LLaMA-2     |              0.76               |                     0.17          |
>
> |     LLaVA-1.5   |              0.67               |                 0.08               |
>
> |     CLIP-ViT + LLaMA-2 (ours, no Q-EE)  | 0.88   |   0.29    |
>
> |    CLIP-ViT + LLaMA-2 + Q-EE (ours)     |   0.90  |   0.31  |
>
> Several observations directly address the concern:
>
> **Image-only vs VLM: LLaVA-1.5**, which ingests the raw image and text jointly, improves over CLIP-ViT by only +0.08 F1 (0.67 vs 0.59). This shows that simply passing the picture through a VLM does not, by itself, yield the large gains we report.
>
>
> **Our fusion vs VLM: Our CLIP-ViT + LLaMA-2** fusion (without Q-EE) reaches 0.88 F1, which is +0.21 points higher than LLaVA-1.5 (0.67), even though all backbone encoders are kept frozen. This indicates that the proposed fusion mechanism over the embedding spaces captures complementary information beyond what a standard VLM architecture extracts from pixels and text.
>
>
> **Effect of Q-EE**: Adding Q-EE further improves performance to 0.90 F1 (+0.31 over CLIP-ViT and +0.23 over LLaVA-1.5), suggesting that the quantum-inspired alignment is not just “using the photo” but provides an additional structured way of integrating visual and textual evidence.
>
> We will clarified this discussion in the revised manuscript and explicitly state that our claim is not that “any embedding is superior to ingesting pixels”.
>
> ###############################################
>
> **Q2**: I would have liked to see how the accuracy changes if the dataset is added to a larger meme dataset, showing a more challenging, yet more realistic setting, in which this type of meme is just one type of abusive memes to be detected and taken down.
>
> **Response:**
> We conducted additional experiments where DACAM was integrated into GOAT-Bench **[a]** as a sixth harm category and evaluated using MM-CAD (LLaMA-2 7B + CLIP-ViT + Q-EE). The results show that the model generalizes reasonably well beyond the DACAM domain. Specifically, we **obtained: F1-Score (Child-Abuse): 0.83, F1-Score (Other Abuses/Non-Child): 0.83, and Macro-F1: 0.81. While the Macro-F1 on DACAM alone was 0.90**, indicating a drop of about 9 points, the performance remains comparable and demonstrates meaningful cross-domain robustness.
>
> Failure cases mostly involve political sarcasm or racial stereotypes—categories absent from DACAM.
>
> **[a]** : **GOAT-Bench** is a recent multimodal benchmark designed to evaluate models on real-world online harassment, toxicity, and hateful content expressed through memes. **The dataset covers five major harm types—Hateful, Offensive, Aggressive, Toxic, and Benign**—each containing memes collected from real platforms such as Reddit, Twitter, and public meme archives.
>
> ##################################################################
>
> **Q3**: In the context of ethics, I would like the authors to discuss more of how they see their work being used.
>
> **Response**: We appreciate this important point. We will expand the ethics section to clarify intended uses: MMCAD is designed to support child-safety organizations—such as NGOs, hotlines, and moderation teams—by helping flag potentially harmful child-abusive memes. It functions strictly as a decision-support tool, not an autonomous moderation system, and may also aid educational awareness efforts.
>
> To **prevent misuse, MMCAD is content-focused: it flags harmful memes, not individuals**. The DACAM dataset will be released only **under ethical-use and academic-research agreements, and all generated explanations are intentionally contextual rather than accusatory**.
>
> We will integrate these clarifications into a dedicated subsection on Ethical Use, Limitations, and Responsible Deployment in the revised manuscript

---

> > ### Author Response · Authors · 2025-11-20
> > **Review Request (BbgV)**
> >
> > Dear Reviewer BbgV
> >
> > We submitted our responses to your reviews two days ago. We would be very grateful if you could kindly take a look at them at your convenience and let us know if any further clarification is needed. This will help us address any follow-up questions in time during the discussion period.
> >
> > Thank you again for your time and thoughtful feedback.

---

### Note · Authors · 2026-04-23

I have read and agree with the venue's withdrawal policy on behalf of myself and my co-authors.

---

### Meta-Review · Area_Chair_v7qT · 2026-01-07

**Summary:**

This submission introduces DACAM, claimed to be the first dataset dedicated to child-abusive memes, and proposes MM-CAD, a multimodal detect-and-explain framework combining CLIP, OCR, LLMs, and a quantum-inspired embedding enhancement (Q-EE). The topic is socially important and underexplored. All three reviewers rated the paper marginally below acceptance but explicitly stated they would not mind if it were accepted, contingent on addressing concerns around validation, generalization, methodological clarity, and clarification on ethics.

The core concerns of reviewers seem to be addressed in substance, with the remaining risk tied to whether promised revisions are clearly integrated into the final manuscript.
This AC cannot see if the paper has been revised as there's no marked up version of the revised paper.
Hence at this stage, I cannot recommend this paper for acceptance.

**Reviewer Concerns:**

Major Concerns:
1. Insufficient Experimental Validation of the Proposed Method
2.  Lack of Generalization Evidence Beyond DACAM
3. Weak Justification of the Quantum-Inspired Embedding Enhancement (Q-EE)
4. Insufficient Justification around Human Evaluation of Explanations
5. Ethics, Dataset Governance, and Responsible Use

The above are addressed on surface level in the rebuttal but the paper has not been revised.

Further, the paper attempts to contribute both a new dataset and a novel detection framework, but neither aspect is developed to the depth typically expected for a full research-track paper. Reviewers noted that the work may be better suited to a dataset-focused contribution unless the methodological claims are substantially strengthened.

**Reviewer Scores:**

They may have changed or increased the scores if the manuscript were revised.

---

### Decision · Program_Chairs · 2026-01-26

Reject